# Postmastectomy radiotherapy in pN1 breast cancer: Survival outcomes and prognostic factors from a single-institution cohort

**Raksha M. Narasimhan**[1]*, **Aren Singh Saini**[1], **Kayla Samimi**[1], **Ifeanyichukwu Ogobuiro**[1,2], **Xuesen Zhao**[2], **Sunwoo Han**[3], **Cristiane Takita**[1,2], **Crystal S. Taswell**[1,2]

**1** Department of Medical Education, University of Miami Miller School of Medicine, Miami, Florida, United States of America, **2** Sylvester Comprehensive Cancer Center, University of Miami, Miami, Florida, United States of America, **3** Department of Biostatistics, Robert Stempel College of Public Health and Social Work, Florida International University, Miami, Florida, United States of America

* rmn58@med.miami.edu

## Abstract

### Purpose/Objectives

The role of postmastectomy radiotherapy (PMRT) in patients with pathologic N1 (pN1) breast cancer, including triple-negative breast cancer (TNBC), remains controversial in the era of modern systemic therapy. We evaluated the association between PMRT and recurrence-free survival (RFS) and overall survival (OS) and identified prognostic factors in a contemporary single-institution pN1 cohort.

### Materials/Methods

We retrospectively reviewed female patients with pT1-2N1M0 breast cancer treated with mastectomy between 1999 and 2023. RFS and OS were estimated using Kaplan-Meier methods and compared by PMRT status with log-rank testing. Univariable Cox proportional hazards models assessed associations between clinical factors—including tumor laterality, receptor subtype (TNBC vs non-TNBC), nodal burden, and adjuvant therapies—and survival outcomes, with subgroup analyses by PMRT status and receptor subtype.

### Results

Fifty-seven patients were included; 22 (38.6%) received PMRT. With a median follow-up of 85 months, PMRT was not associated with improved RFS (median 133 vs 120 months; p = 0.256) or OS (not reached vs 195 months; p = 0.154). Hormone therapy was significantly associated with improved RFS (HR 0.43; p = 0.026) and OS (HR 0.13; p = 0.003), while having 2–3 positive lymph nodes predicted worse RFS (HR 2.86; p = 0.007). No significant differential benefit from PMRT was observed in patients with TNBC or non-TNBC disease.

**Data availability statement:** Data supporting the findings of this study are not publicly available due to institutional and ethical restrictions related to patient confidentiality. The data are available upon reasonable request from the University of Miami Human Subjects Research Office, which serves as the institutional oversight body for this study. Requests for data access may be directed to: Human Subjects Research Office (HSRO) University of Miami Email: hsro@miami.edu Phone: 305-243-3195 Website: www.hsro.miami.edu Study reference: HSRO Study Number: 20090856 Study Title: Collection of Characteristics and Outcome of Patients Treated for Breast Cancer by Physicians in the Department of Radiation Oncology IRB Approval Date: December 1, 2009 Data access is subject to institutional review and approval, and may require a data use agreement in accordance with University of Miami policies.

**Funding:** The author(s) received no specific funding for this work.

**Competing interests:** The authors have declared that no competing interests exist.

## Conclusions

PMRT was not associated with a survival benefit in this pN1 cohort, including patients with TNBC. Interpretation is limited by modest sample size and statistical power. Outcomes appeared driven by tumor biology, nodal burden, and systemic therapy, supporting individualized PMRT decision-making.

## Introduction

Postmastectomy radiation therapy (PMRT) is a well-established component of adjuvant treatment for high-risk breast cancer, significantly improving locoregional control and survival in patients with extensive nodal involvement [1]. The Danish 82b and 82c studies demonstrated that adding PMRT to mastectomy and systemic therapy markedly reduces locoregional recurrence and improves long-term survival in node-positive disease, firmly establishing PMRT as standard of care for patients with highest risk features [1,2]. A subsequent meta-analysis by the Early Breast Cancer Trialists' Collaborative Group (EBCTCG) further confirmed that PMRT confers significant reductions in recurrence and breast cancer mortality, even in patients with limited nodal disease [3].

Despite these data, the use of PMRT in patients with only one to three positive axillary lymph nodes (pN1) remains one of the most debated issues in breast cancer management. Patients with node-negative tumors (pN0) generally do not require PMRT due to very low locoregional recurrence risk, whereas those with four or more involved nodes (N2+) clearly benefit and are routinely treated; the pN1 population lies between these extremes [4]. Notably, the EBCTCG meta-analysis found that PMRT in women with 1–3 positive nodes yielded an absolute ~8% reduction in 20-year breast cancer mortality [3].

As many patients in those trials lacked contemporary chemotherapy, HER2-targeted therapy, and thorough axillary dissection, it is uncertain whether the same magnitude of benefit applies to today's pN1 patients [5]. Several tumor and patient factors have been identified that influence locoregional recurrence risk in pN1 disease and thus may modulate the absolute benefit of PMRT. For example, triple-negative or HER2-positive breast cancers and a higher number of positive nodes are associated with greater baseline recurrence risk and potentially greater benefit from PMRT, whereas patients with small, low-grade, hormone receptor-positive tumors and minimal nodal involvement have a lower risk of recurrence [6,7]. Tumor laterality is another consideration: left-sided cases historically carried higher cardiac toxicity from radiotherapy, though modern techniques have substantially mitigated this risk [8]. Accordingly, the National Comprehensive Cancer Network (NCCN) recommends considering PMRT for patients with 1–3 positive nodes, particularly if high-risk features are present, with omission for node-negative disease [4]. Similarly, a 2016 consensus update by the American Society of Clinical Oncology (ASCO), the American Society for Radiation Oncology (ASTRO), and the Society of Surgical Oncology (SSO) concluded that PMRT significantly lowers recurrence and breast cancer

mortality in pN1 overall, but emphasized individualized decision-making, noting that certain low-risk pN1 patients may reasonably be spared from PMRT to avoid undue toxicity [9].

The Selective Use of Postmastectomy Radiotherapy (SUPREMO) trial prospectively evaluated PMRT in intermediate-risk patients by randomizing women with T1-2N1 breast cancer or with high-risk N0 disease to PMRT versus no radiation after mastectomy. Its 10-year results showed no significant improvement in overall survival with PMRT, with approximately 81% 10-year survival in both arms. Although radiotherapy roughly halved the rate of chest wall recurrence, the absolute reduction in locoregional recurrence was under 2%, with no significant impact on distant metastasis or breast cancer mortality [10]. These results underscore that many pN1 patients may not gain a meaningful survival benefit from PMRT in the era of effective hormone therapy, reinforcing the need to better identify which patients truly benefit. This study was therefore undertaken to further clarify the role of PMRT in pN1 breast cancer and to identify prognostic risk factors for survival outcomes in this intermediate-risk population.

## Materials and methods

We conducted a single-institution retrospective cohort study of female patients diagnosed with pathologic T1N1M0 or T2N1M0 breast cancer who underwent mastectomy between 1999 and 2023. Eligible patients had 1–3 positive axillary lymph nodes on final pathology, no clinical or radiographic evidence of distant metastasis at diagnosis (M0), and documentation of treatment and follow-up. Patients with bilateral breast cancer and prior invasive breast malignancy were excluded.

Clinical and pathological data were abstracted from electronic medical records by IRB-authorized institutional breast cancer registry personnel on November 30, 2023. Collected variables included age at diagnosis, self-reported race/ethnicity, tumor size and laterality, histology, estrogen receptor (ER) status, progesterone receptor (PR) status, HER2 status, and use of adjuvant therapies including chemotherapy, endocrine therapy, HER2-directed therapy, and receipt of postmastectomy radiation therapy (PMRT). The variables were selected based on clinical relevance and prior literature [11–13].

The primary outcomes were recurrence-free survival (RFS) and overall survival (OS). RFS was defined as elapsed time from diagnosis to recurrence, death from any cause, or last follow-up. OS was defined as elapsed time from diagnosis to death or last follow-up.

Descriptive statistics were used to summarize baseline characteristics. Continuous variables were summarized with mean, median, standard deviation (SD) and interquartile range (IQR), and two independent groups were compared using Wilcoxon rank-sum test. Categorical variables were tabulated with frequency and percentage, and two groups were compared using Chi-squared test or Fisher's exact test. Kaplan-Meier method was employed to estimate RFS and OS, and groups were compared using the log-rank test. Cox proportional hazards regression models were used to evaluate associations between prognostic factors and RFS and OS. Analyses were applied for the whole cohort and stratified by receipt of postmastectomy radiotherapy (PMRT): PMRT and non-PMRT cohorts. To explore the effect of PMRT by triple-negative receptor status, we also conducted the corresponding subset analyses. Across all the regression modeling, multivariable analyses were not performed due to the small sample size. Statistical significance was defined as a two-sided p-value <0.05. All analyses were performed using SAS version 9.4.

This study was conducted using data from an institutional breast cancer registry approved by the University of Miami Institutional Review Board (IRB #20090856). The registry includes both retrospectively and prospectively collected clinical data from patients treated for breast cancer within the Department of Radiation Oncology at the University of Miami. For the purposes of this analysis, only retrospectively collected data were used. The data was accessed for research purposes on November 30th, 2023 by authorized registry personnel in accordance with the approved protocol. In accordance with the approved protocol, retrospective data collection was conducted under a waiver of informed consent and a waiver of HIPAA authorization. Registry personnel were not involved in study design, analysis, interpretation, or manuscript preparation.

Authorized registry personnel had access to identifiable patient information during data abstraction; all data were deidentified prior to transfer to the study authors, who did not have access to identifiable information during or after analysis. No direct interaction with patients occurred, and no patients were contacted for this study. Data were abstracted from existing medical records and entered into a secure, HIPAA-compliant REDCap database accessible only to IRB-approved study personnel.

## Results

A total of 57 patients with pT1-2N1M0 breast cancer who underwent mastectomy were included in the study, of whom 22 (38.6%) received postmastectomy radiation therapy (PMRT) and 35 (61.4%) did not. The cohort was predominantly hormone receptor-positive and HER2-negative (83.4%), with triple-negative disease comprising 16.7%. Most patients had a single positive lymph node (76.5%), and the majority received chemotherapy (63.6%) and hormone therapy (66.1%). Baseline demographic and clinical characteristics were compared between the two groups, with no statistically significant differences observed in all variables (p > .05). Of the 54 patients with reported receptor status, the vast majority were hormone receptor-positive and HER2-negative (83.4% overall; 77.2% without PMRT vs. 94.7% with PMRT), while triple-negative tumors accounted for 16.7% (22.9% without PMRT vs. 5.3% with PMRT). Receptor subtype distribution did not differ significantly between treatment groups (Table 1). For the whole study cohort, median follow-up from diagnosis was 85 months (IQR: 43–156).

### Recurrence-free and overall survival

Of the 57 patients, there were 31 recurrence events—5 local, 2 nodal, and 24 systemic recurrences—and 12 deaths (Table 1). For the whole cohort, median RFS and OS were 133 months (95% CI: 72–155) and not estimable (NE), respectively (Fig 1). When stratified by receipt of PMRT, there was no statistically significant difference in RFS or OS between the PMRT and non-PMRT groups. Median RFS was 133 months (95% CI: 72-NE) in the PMRT group versus 120 months (95% CI: 33–155) in the non-PMRT group (log-rank p = 0.256). Median OS was not reached in the PMRT group and was 195 months (95% CI: 124-NE) in the non-PMRT group (log-rank p = 0.154) (Fig 2).

### Prognostic factors of recurrence-free survival

In univariable Cox regression analyses across the entire cohort (Table 2), receipt of hormone therapy was significantly associated with improved RFS (HR = 0.43, 95% CI: 0.21–0.90, p = 0.026). In contrast, having two or three positive lymph nodes was associated with worse RFS (HR = 2.86, 95% CI: 1.34–6.11, p = 0.007), indicating nearly a threefold increased risk of recurrence compared to a single positive node. No other clinicopathologic variables, including age, tumor size, receptor subtype, or receipt of PMRT, were significantly associated with RFS (Table 2). There were no statistically significant differences in RFS between the PMRT and non-PMRT group (HR = 0.65, 95% CI: 0.30–1.38, p = 0.263) (Fig 2).

In subset analyses with patients who did not receive PMRT, univariable Cox models identified left-sided tumor laterality as a significant predictor of improved RFS compared to right-sided disease (HR = 0.25, 95% CI: 0.09–0.73, p = 0.011). A higher nodal burden (2–3 nodes) was also associated with increased recurrence risk compared to single node in this subgroup (HR = 2.89, 95% CI: 1.15–7.27, p = 0.024). In the subset analyses with the PMRT group, none of prognostic factors was identified as significant predictors, likely reflecting the limited sample size and number of events (Table 2).

### Prognostic factors of overall survival

In the analyses of OS for the entire cohort (Table 3), hormone therapy was significantly associated with improved survival (HR = 0.13, 95% CI: 0.04–0.50, p = 0.003), representing an approximately 87% reduction in mortality risk. ER + PR + HER2-receptor status was also significantly associated with longer OS relative to triple-negative disease (HR = 0.25, 95% CI:

**Table 1. Demographics & Clinical Variables.**

| | Total(N = 57) N (%) | Non-PMRT(N = 35) N (%) | PMRT(N = 22) N (%) | P-value |
|---|---|---|---|---|
| **Age at Diagnosis** | | | | |
| Mean (SD) | 57.3 (12.5) | 58.5 (13) | 55.5 (11.7) | 0.3962[1] |
| Median (IQR) | 57 (50, 63) | 57.0 (50, 65) | 56.0 (44, 63) | |
| Min, Max | 29, 88 | 29, 88 | 37, 75 | |
| **Race** | | | | 1.0000[2] |
| Black | 8 (14.0) | 5 (14.3) | 3 (13.6) | |
| White | 49 (86.0) | 30 (85.7) | 19 (86.4) | |
| **Ethnicity** | | | | 0.9527[3] |
| Non-Hispanic | 36 (63.2) | 22 (62.9) | 14 (63.6) | |
| Hispanic | 21 (36.8) | 13 (37.1) | 8 (36.4) | |
| **Laterality** | | | | 0.3261[2] |
| Right | 27 (47.4) | 14 (40.0) | 13 (59.1) | |
| Left | 29 (50.9) | 20 (57.1) | 9 (40.9) | |
| Paired Site | 1 (1.8) | 1 (2.9) | 0 (0.0) | |
| **Cancer Type**, n (%) | 49 (100) | 32 (100) | 17 (100) | 0.1139[2] |
| IDC | 45 (91.8) | 31 (96.9) | 14 (82.4) | |
| ILC | 4 (8.2) | 1 (3.1) | 3 (17.6) | |
| **Chemotherapy**, n (%) | 55 (100) | 34 (100) | 21 (100) | 0.5764[2] |
| None | 20 (36.4) | 14 (41.2) | 6 (28.6) | |
| NOS | 4 (7.3) | 3 (8.8) | 1 (4.8) | |
| Single Agent | 3 (5.5) | 1 (2.9) | 2 (9.5) | |
| Multiple Agents | 28 (50.9) | 16 (47.1) | 12 (57.1) | |
| **Multiple Chemotherapy**, n (%) | 55 (100) | 34 (100) | 21 (100) | 0.4674[3] |
| None/NOS/Single | 27 (49.1) | 18 (52.9) | 9 (42.9) | |
| Multiple Agents | 28 (50.9) | 16 (47.1) | 12 (57.1) | |
| **Hormone Therapy**, n (%) | 56 (100) | 34 (100) | 22 (100) | 0.0815[2] |
| No | 19 (33.9) | 15 (44.1) | 4 (18.2) | |
| Yes | 37 (66.1) | 19 (55.9) | 18 (81.8) | |
| **Immunotherapy** | | | | 0.2763[2] |
| No | 54 (94.7) | 32 (91.4) | 22 (100.0) | |
| Yes | 3 (5.3) | 3 (8.6) | 0 (0.0) | |
| **Pathologic Tumor Size**, n (%) | 56 (100) | 35 (100) | 21 (100) | 0.8922[2] |
| p1a | 1 (1.8) | 1 (2.9) | 0 (0.0) | |
| p1b | 4 (7.1) | 3 (8.6) | 1 (4.8) | |
| p1c | 16 (28.6) | 9 (25.7) | 7 (33.3) | |
| p2 | 35 (62.5) | 22 (62.9) | 13 (61.9) | |
| **Surgical Pathologic Stage** | | | | 0.4209[2] |
| 1a | 2 (3.5) | 1 (2.9) | 1 (4.5) | |
| 1b | 6 (10.5) | 2 (5.7) | 4 (18.2) | |
| 1c | 21 (36.8) | 12 (34.3) | 9 (40.9) | |
| 2a | 26 (45.6) | 18 (51.4) | 8 (36.4) | |
| 2b | 2 (3.5) | 2 (5.7) | 0 (0.0) | |
| **Receptor Status**, n (%) | 54 (100) | 35 (100) | 19 (100) | 0.1075[2] |
| Triple Negative | 9 (16.7) | 8 (22.9) | 1 (5.3) | |
| ER + PR + HER2- | 42 (77.8) | 24 (68.6) | 18 (94.7) | |

*(Continued)*

**Table 1.** (Continued)

| | Total(N = 57) N (%) | Non-PMRT(N = 35) N (%) | PMRT(N = 22) N (%) | P-value |
|---|---|---|---|---|
| ER + PR-HER2- | 3 (5.6) | 3 (8.6) | 0 (0.0%) | |
| **Number of Positive Lymph Nodes, n (%)** | 51 (100) | 32 (100) | 19 (100) | 0.3317[2] |
| 1 | 39 (76.5) | 24 (75.0) | 15 (78.9) | |
| 2 | 9 (17.6) | 7 (21.9) | 2 (10.5) | |
| 3 | 3 (5.9) | 1 (3.1) | 2 (10.5) | |
| **Recurrence Occurred** | | | | 0.424[3] |
| No | 26 (45.6) | 14 (40.0) | 12 (54.5) | |
| Yes | 31 (54.4) | 21 (60.0) | 10 (45.5) | |
| Local | 5 (8.8) | 4 (11.4) | 1 (4.5) | |
| Nodal | 2 (3.5) | 2 (5.7) | 0 (0.0) | |
| Systemic | 24 (42.1) | 15 (42.9) | 9 (40.9) | |
| **Vital Status** | | | | 0.153[3] |
| Dead | 12 (21.1) | 10 (28.6) | 2 (9.1) | |
| Alive | 45 (78.9) | 25 (71.4) | 20 (90.9) | |

[1]Equal variance two sample t-test; [2]Fisher Exact p-value; [3]Chi-Square p-value; SD: Standard deviation; IQR: Interquartile Range.

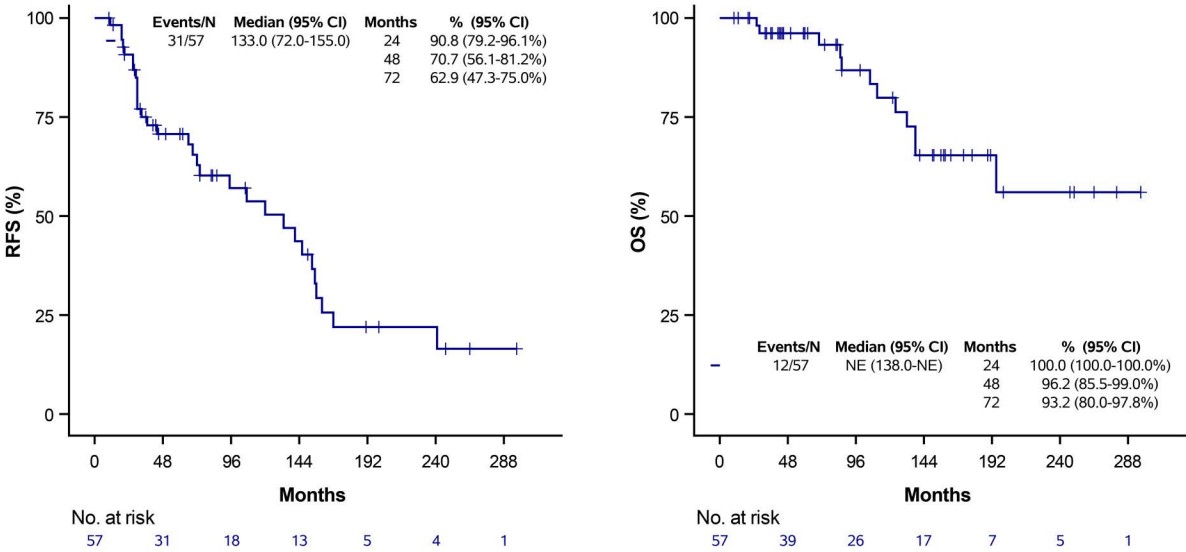

**Fig 1. Kaplan-Meier estimates of RFS and OS for all patients.** Tick marks denote censored observations.

0.08–0.80, p = 0.019) (Table 3). There were no statistically significant differences in OS between the PMRT group and non-PMRT group, and other clinical variables were not significantly associated with survival (HR = 0.33, 95% CI: 0.07–1.51, p = 0.154) (Fig 2).

Among patients who did not receive PMRT, the subset analyses determined that left-sided tumor laterality (HR = 0.22, 95% CI: 0.05–0.89, p = 0.033) and receipt of hormone therapy (HR = 0.15, 95% CI: 0.03–0.70, p = 0.016) were significantly

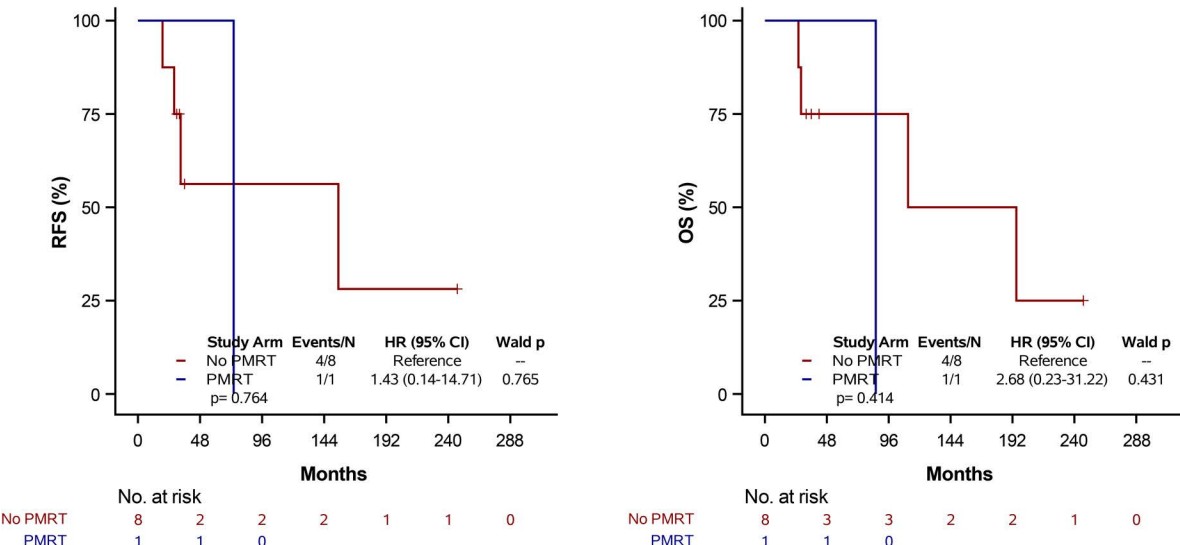

**Fig 2. Kaplan-Meier estimates of RFS and OS stratified by PMRT status.** Tick marks denote censored observations. Survival comparisons were performed using the log-rank test.

associated with improved OS. Due to the limited number of death events (n = 2) in the PMRT group, the corresponding subset analysis was not performed (Table 3).

### Receptor status analysis

Kaplan-Meier analyses were performed stratified by receptor subtype as triple-negative versus non-triple-negative (Figs 3 and 4). Among patients with triple-negative breast cancer, no statistically significant difference in RFS or OS was observed between PMRT and non-PMRT groups (log-rank p > .05) (Fig 3). Similarly, among patients with non-triple-negative receptor status, PMRT was not associated with a statistically significant improvement in RFS or OS (log-rank p > .05) (Fig 4).

In subset analyses of patients who did not have triple negative receptor (S1 Table), univariable Cox model identified hormone therapy as a significant predictor of improved RFS compared to right-sided disease (HR = 0.21, 95% CI: 0.08–0.57, p = 0.002). Having 2–3 lymph nodes was also associated with an increased recurrence risk compared to having a single node in this subgroup (HR = 2.53, 95% CI: 1.07–5.96, p = 0.034). However, no prognostic factors were identified as significant predictors in the subset analyses with triple negative receptor group (S1 Table).

For patients who did not have triple negative receptor, age (1-year increase HR = 1.14, 95% CI: 1.01–1.29, p = 0.039) and hormone therapy (HR = 0.12, 95% CI: 0.03–0.56, p = 0.007) were identified as significant factors for OS. In patients who have the triple negative receptor type, no prognostic factors were significantly associated with OS (S2 Table).

### Discussion

Our study demonstrates no statistically significant difference in OS and RFS when comparing PMRT and non-PMRT groups with pN1 breast cancer, suggesting that PMRT is not necessary for pN1 breast cancer patients. Many studies have not shown improvement in OS in N1 breast cancer patients who receive PMRT, similar to our findings [14–16]. While our findings are consistent with prior retrospective and observational studies, they provide contemporary validation—in the modern treatment era, advances in systemic therapy may attenuate the absolute benefit of PMRT. Taken in this context,

**Table 2. Cox Regression Analyses Assessing Risk Factors for RFS.**

| Variable | Category | HR (95%CI) | P |
|---|---|---|---|
| *Univariable (n = 57, 31 events)* | | | |
| Age | 1-unit increase | 1.02 (0.99, 1.06) | 0.193 |
| | >60 vs ≤ 60 (ref) | 1.27 (0.59, 2.74) | 0.536 |
| Race | White vs Black (ref) | 2.01 (0.48, 8.52) | 0.342 |
| Ethnicity | Hispanic vs Non-Hispanic (ref) | 0.79 (0.37, 1.68) | 0.535 |
| Laterality (n = 56, 30 events) | Left vs Right (ref) | 0.65 (0.31, 1.34) | 0.243 |
| Cancer Type (n = 49, 26 events) | ILC vs IDC (ref) | 0.83 (0.19, 3.53) | 0.796 |
| Chemotherapy (n = 55, 30 events) | NOS/Single/Multiple vs None (ref) | 0.59 (0.26, 1.31) | 0.194 |
| Multiple Chemotherapy (n = 55, 30 events) | Multiple Agents vs None/NOS/Single Agent (ref) | 0.72 (0.35, 1.47) | 0.365 |
| Hormone Therapy (n = 56, 30 events) | Yes vs No (ref) | 0.43 (0.21, 0.90) | **0.026** |
| Immunotherapy | Yes vs No (ref) | 1.87 (0.56, 6.23) | 0.308 |
| Tumor Size (n = 56, 30 events) | p2 vs p1Mi/p1a/p1b/p1c (ref) | 1.30 (0.61, 2.76) | 0.494 |
| Surgical Pathological Stage | 2a/2b vs 1a/1b/1c (ref) | 1.50 (0.73, 3.07) | 0.274 |
| Receptor Status (n = 54, 29 events) | ER + PR + HER2- vs Triple Negative (ref) | 0.90 (0.34, 2.39) | 0.839 |
| | ER + PR-HER2- vs Triple Negative (ref) | 0.38 (0.04, 3.26) | 0.376 |
| Number of Positive Lymph Nodes | 2-3 vs 1 (ref) | 2.86 (1.34, 6.11) | **0.007** |
| *Subset univariable analyses:1. PMRT (n = 22, 10 events)* | | | |
| Age | 1-unit increase | 1.03 (0.97, 1.10) | 0.335 |
| | >60 vs ≤ 60 (ref) | 2.66 (0.70, 10.18) | 0.154 |
| Race | White vs Black (ref) | N/A | N/A |
| Ethnicity | Hispanic vs Non-Hispanic (ref) | 0.65 (0.17, 2.55) | 0.540 |
| Laterality | Left vs Right (ref) | 1.10 (0.31, 3.94) | 0.884 |
| Cancer Type (n = 17, 7 events) | ILC vs IDC (ref) | 0.70 (0.08, 6.01) | 0.745 |
| Chemotherapy (n = 21, 10 events) | NOS/Single/Multiple vs None (ref) | 0.19 (0.04, 1.01) | 0.052 |
| Multiple Chemotherapy (n = 21, 10 events) | Multiple Agents vs None/NOS/Single Agent (ref) | 0.41 (0.12, 1.42) | 0.159 |
| Hormone Therapy | Yes vs No (ref) | 0.41 (0.10, 1.72) | 0.222 |
| Immunotherapy | Yes vs No (ref) | N/A | N/A |
| Tumor Size (n = 21, 9 events) | p2 vs p1Mi/p1a/p1b/p1c | 2.15 (0.44, 10.54) | 0.345 |
| Surgical Pathological Stage | 2a/2b vs 1a/1b/1c (ref) | 1.93 (0.53, 7.06) | 0.322 |
| Receptor Status (n = 19, 8 events) | ER + PR + HER2- vs Triple Negative (ref) | 0.34 (0.04, 3.12) | 0.342 |
| | ER + PR-HER2- vs Triple Negative (ref) | N/A | N/A |
| Number of Positive Lymph Nodes (n = 19, 9 events) | 2-3 vs 1 (ref) | 2.83 (0.72, 11.08) | 0.135 |
| *2. Non-PMRT (n = 35, 21 events)* | | | |
| Age | 1-unit increase | 1.01 (0.97, 1.05) | 0.575 |
| | >60 vs ≤ 60 (ref) | 0.83 (0.30, 2.27) | 0.714 |

*(Continued)*

**Table 2.** (Continued)

| Variable | Category | HR (95%CI) | P |
|---|---|---|---|
| Race | White vs Black (ref) | 1.70 (0.39, 7.39) | 0.477 |
| Ethnicity | Hispanic vs Non-Hispanic (ref) | 0.81 (0.32, 2.01) | 0.643 |
| Laterality (n = 34, 20 events) | Left vs Right (ref) | 0.25 (0.09, 0.73) | **0.011** |
| Cancer Type (n = 32, 19 events) | ILC vs IDC (ref) | 1.87 (0.24, 14.57) | 0.550 |
| Chemotherapy (n = 34, 20 events) | NOS/Single/Multiple vs None (ref) | 0.89 (0.33, 2.37) | 0.813 |
| Multiple Chemotherapy (n = 34, 20 events) | Multiple Agents vs None/NOS/Single Agent (ref) | 1.00 (0.41, 2.43) | 0.997 |
| Hormone Therapy (n = 34, 20 events) | Yes vs No (ref) | 0.49 (0.20, 1.20) | 0.119 |
| Immunotherapy | Yes vs No (ref) | 1.58 (0.45, 5.46) | 0.473 |
| Tumor Size | p2 vs p1Mi/p1a/p1b/p1c (ref) | 1.13 (0.47, 2.72) | 0.778 |
| Surgical Pathological Stage | 2a/2b vs 1a/1b/1c (ref) | 1.26 (0.53, 3.01) | 0.602 |
| Receptor Status | ER + PR + HER2- vs Triple Negative (ref) | 1.40 (0.47, 4.23) | 0.548 |
| | ER + PR-HER2- vs Triple Negative (ref) | 0.43 (0.05, 3.86) | 0.448 |
| Number of Positive Lymph Nodes (n = 32, 19 events) | 2-3 vs 1 (ref) | 2.89 (1.15, 7.27) | **0.024** |

HR: Hazard Ratio assessing risk of RFS events for 1-unit increase, or for specific category vs reference; CI: Confidence Interval; P: p-value testing HR = 1 from Wald Test.

N/A: not applicable due to the small number of events in a group.

**Note:** For a variable with missing value, we clarified the evaluable number of participants and events after excluding the missing values.

our results contribute to ongoing efforts to refine risk-adapted, individualized decision-making for pN1 breast cancer patients.

Similar prior studies have shown that PMRT can reduce locoregional recurrence and improve RFS in N1 breast cancer patients when comparing certain subgroups. These subgroups include patients with increased tumor size, younger age, increased number of node involvement, and triple-negative subtype [6,17–19]. Consistent with these claims, involvement of multiple nodes and triple negative disease was associated with decreased survival in our cohort. Though these prognostic factors have previously been established, their presence in this cohort reinforces their continued clinical relevance in contemporary practice.

The literature suggests that PMRT may confer a benefit in pN1 triple negative breast cancer (TNBC) patients given the higher risk of locoregional recurrence in these patients [20,21]. However, exploratory subgroup analyses did not demonstrate a statistically significant differential benefit from PMRT by receptor subtype (Figs 3 and 4). These findings should be interpreted cautiously given the small number of triple-negative cases (n = 9), particularly among patients receiving PMRT.

Notably, the absolute benefit of PMRT in N1 disease may be modest depending on response to systemic chemotherapy and other therapies. For instance, Wang et al. demonstrated a statistically significant improvement in survival parameters for all patients who received PMRT even after receiving systemic chemotherapy. However, for patients with pathologic complete response to systemic therapy, both RFS and OS did not improve after PMRT [22]. This further adds to the nuance and calls for an individualized approach to PMRT for pN1 breast cancer patients. Our study adds to the existing pool of literature to help characterize these cohorts.

**Table 3. Cox Regression Analyses Assessing Risk Factors for OS.**

| Variable | Category | HR (95%CI) | P |
|---|---|---|---|
| **Univariable (n = 57, 12 events)** | | | |
| Age | 1-unit increase | 1.05 (0.99, 1.11) | 0.126 |
| | >60 vs ≤60 (ref) | 1.77 (0.56, 5.66) | 0.333 |
| Race | White vs Black (ref) | 1.07 (0.14, 8.48) | 0.946 |
| Ethnicity | Hispanic vs Non-Hispanic (ref) | 0.98 (0.29, 3.29) | 0.974 |
| Laterality (n = 56, 12 events) | Left vs Right (ref) | 0.58 (0.18, 1.83) | 0.354 |
| Cancer Type (n = 49, 10 events) | ILC vs IDC (ref) | 1.11 (0.14, 8.90) | 0.921 |
| Chemotherapy (n = 55, 12 events) | NOS/Single/Multiple vs None (ref) | 0.75 (0.20, 2.83) | 0.671 |
| Multiple Chemotherapy (n = 55, 12 events) | Multiple Agents vs None/NOS/Single Agent (ref) | 0.94 (0.30, 2.98) | 0.922 |
| Hormone Therapy (n = 56, 12 events) | Yes vs No (ref) | 0.13 (0.04, 0.50) | **0.003** |
| Immunotherapy | Yes vs No (ref) | 1.53 (0.19, 11.98) | 0.687 |
| Tumor Size (n = 56, 12 events) | p2 vs p1Mi/p1a/p1b/p1c (ref) | 1.56 (0.47, 5.19) | 0.469 |
| Surgical Pathological Stage | 2a/2b vs 1a/1b/1c (ref) | 2.04 (0.61, 6.79) | 0.244 |
| Receptor Status (n = 54, 12 events) | ER+PR+HER2- vs Triple Negative (ref) | 0.25 (0.08, 0.80) | **0.019** |
| | ER+PR-HER2- vs Triple Negative (ref) | N/A | N/A |
| Number of Positive Lymph Nodes (n = 51, 12 events) | 2-3 vs 1 (ref) | 1.82 (0.57, 5.78) | 0.313 |
| Recurrence Occurred | Yes vs No (ref) | 2.56 (0.55, 11.93) | 0.231 |
| Recurrence Type | Local/Nodal vs No Recurrence (ref) | 4.34 (0.69, 27.25) | 0.117 |
| | Systemic vs No Recurrence (ref) | 2.21 (0.45, 10.83) | 0.329 |
| **Subset univariable Analyses:1. PMRT (n = 22, 2 events) — not performed due to only 2 death events** | | | |
| **2. Non-PMRT (n = 35, 10 events)** | | | |
| Age | 1-unit increase | 1.05 (0.98, 1.12) | 0.146 |
| | >60 vs ≤60 (ref) | 1.71 (0.48, 6.10) | 0.407 |
| Race | White vs Black (ref) | 1.30 (0.16, 10.42) | 0.805 |
| Ethnicity | Hispanic vs Non-Hispanic (ref) | 0.87 (0.22, 3.39) | 0.842 |
| Laterality (n = 34, 10 events) | Left vs Right (ref) | 0.22 (0.05, 0.89) | **0.033** |
| Cancer Type (n = 32, 10 events) | ILC vs IDC (ref) | N/A | N/A |
| Chemotherapy (n = 34, 10 events) | NOS/Single/Multiple vs None (ref) | 0.87 (0.22, 3.42) | 0.845 |
| Multiple Chemotherapy (n = 34, 10 events) | Multiple Agents vs None/NOS/Single Agent (ref) | 0.87 (0.25, 3.01) | 0.823 |
| Hormone Therapy (n = 34, 10 events) | Yes vs No (ref) | 0.15 (0.03, 0.70) | **0.016** |
| Immunotherapy | Yes vs No (ref) | 1.23 (0.15, 9.88) | 0.846 |
| Tumor Size | p2 vs p1Mi/p1a/p1b/p1c (ref) | 2.03 (0.52, 7.87) | 0.306 |

*(Continued)*

**Table 3.** (Continued)

| Variable | Category | HR (95%CI) | P |
|---|---|---|---|
| Surgical Pathological Stage | 2a/2b vs 1a/1b/1c (ref) | 2.08 (0.54, 8.05) | 0.290 |
| Receptor Status | ER+PR+HER2- vs Triple Negative (ref) | 0.43 (0.12, 1.58) | 0.204 |
| | ER+PR-HER2- vs Triple Negative (ref) | N/A | N/A |
| Number of Positive Lymph Nodes (n=32, 10 events) | 2-3 vs 1 (ref) | 2.69 (0.76, 9.49) | 0.124 |
| Recurrence Occurred | Yes vs No (ref) | 3.50 (0.43, 28.37) | 0.241 |
| Recurrence Type | Local/Nodal vs No Recurrence (ref) | 5.01 (0.49, 51.51) | 0.175 |
| | Systemic vs No Recurrence (ref) | 3.10 (0.36, 26.34) | 0.300 |

HR: Hazard Ratio assessing risk of death for 1-unit increase, or for specific category vs reference; CI: Confidence Interval; P: p-value testing HR=1 from Wald Test.

N/A: not applicable due to the small number of events in a group.

**Note:** For a variable with missing value, we clarified the evaluable number of participants and events after excluding the missing values.

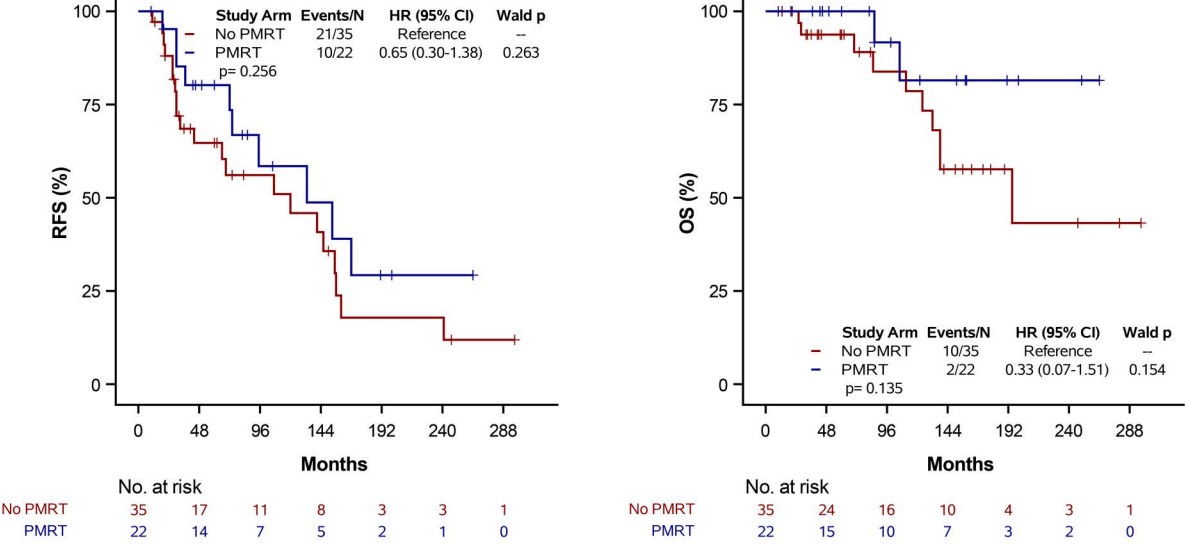

**Fig 3. Kaplan-Meier estimates of RFS and OS stratified by PMRT status among TNBC patients (N=9).** Tick marks denote censored observations. Survival comparisons were performed using the log-rank test.

Of note, stratification tools such as the 21-gene recurrence score (RS) assay have been studied to identify patients who may gain survival benefit from PMRT. A large observational cohort study of T1-2 N1 estrogen receptor-positive breast cancer patients from the National Cancer Database (NCDB) showed that patients with a low RS score had improved overall survival with PMRT [23,24]. Yet a later study by Zhang et al. showed no difference in overall survival regardless of RS [25]. Together with the present findings, these data highlight the heterogeneity of pN1 breast cancer and suggest that molecular and biologic features, including receptor status, may inform risk stratification, though definitive subtype-specific

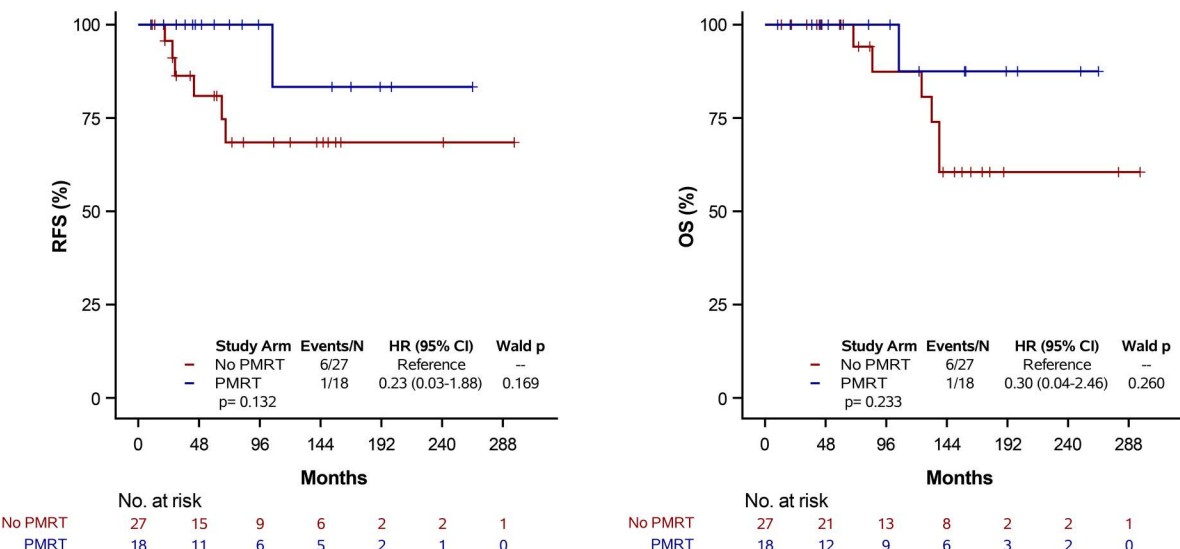

**Fig 4. Kaplan-Meier estimates of RFS and OS stratified by PMRT status among non-TNBC patients (N = 45).** Tick marks denote censored observations. Survival comparisons were performed using the log-rank test.

benefit from PMRT remains unproven. These results highlight that each patient's pathology is unique and highlights the need to develop individualized treatment plans for N1 breast cancer patients.

On subgroup analysis, patients who did not receive PMRT with left-sided tumor laterality had improved RFS and OS compared to those with right-sided tumors. The decreased prognosis in left-sided breast cancer compared to right-sided disease is consistent with outside literature [26]. Improved survival without radiation is also expected, as radiation to the left chest wall may cause long-term damage to nearby structures of the heart and lungs. Radiation therapy in left-sided breast cancer is associated with coronary artery disease, increased 10-year risk of myocardial infarctions, radiation-induced pneumonitis, and other related conditions [8,27–29]. All of these secondary effects worsen mortality. Therefore, this study's finding parallels external findings.

## Limitations

This study is limited by its retrospective design, which introduces selection bias and limits the ability to establish causal relationships. There was variability in systemic therapy regimens including chemotherapy, endocrine therapy, and targeted agents as well as differences in surgical techniques across treating physicians and over time, which may have influenced outcomes. Pathologic and molecular data were occasionally incomplete, with some patients missing HER2 status, Ki67, or other prognostic biomarkers, which could limit risk stratification. Subgroup analyses stratified by receptor subtype were limited by small sample size and event counts and should be considered exploratory. Follow-up duration was heterogeneous and may not be sufficient to capture late toxicities, second malignancies, or long-term survival differences. Residual confounding from unmeasured variables such as comorbidities, adherence to adjuvant systemic therapy, and socioeconomic factors may have impacted both treatment selection and outcomes. This study also did not collect toxicity or quality of life data, which prevents evaluation of the morbidity associated with postmastectomy radiation therapy such as lymphedema, cardiopulmonary toxicity, and other late effects. These limitations highlight the need for larger prospective multi-institutional studies with standardized treatment protocols, comprehensive molecular profiling, and inclusion of patient-reported outcomes to better define the role of postmastectomy radiation therapy in node positive breast cancer.

These factors may limit the generalizability of our findings to broader and more diverse patient populations. Additionally, the study may be underpowered to detect modest but clinically meaningful differences in survival outcomes between treatment groups. While this study does not introduce novel prognostic factors, it contributes to the existing literature by evaluating outcomes in a contemporary, real-world cohort with detailed subgroup analyses, which may help contextualize historical trial data in the setting of modern systemic therapy.

## Conclusions

There remains a need for prospective, randomized trials to clarify the role of PMRT in pN1 breast cancer in the context of modern systemic therapy. Existing evidence is largely retrospective or derived from subgroup analyses, with heterogeneous patient populations and sometimes conflicting results. In our single-institution retrospective cohort, PMRT was not associated with statistically significant differences in RFS or OS, including patients with TNBC. However, these findings should be interpreted cautiously given the modest sample size and limited statistical power. These findings underscore the heterogeneity of pN1 breast cancer and support continued efforts toward personalized, risk-adapted PMRT selection, ideally validated in larger multi-institutional prospective cohorts incorporating contemporary systemic therapy and tumor biology. In addition, they provide context for ongoing debates regarding the role of PMRT in pN1 breast cancer in the modern treatment era and support individualized approaches to treatment selection informed by clinical, pathologic, and biologic risk factors.

## Supporting information

**S1 Table. Cox regression analyses assessing risk factors for RFS by receptor status (TNBC vs. non-TNBC).** HR: Hazard Ratio assessing risk of RFS events for 1-unit increase, or for specific category vs reference; CI: Confidence Interval; P: p-value testing HR = 1 from Wald Test. N/A: not applicable due to the small number of events in a group. Note: For a variable with missing value, we clarified the evaluable number of participants and events after excluding the missing values.
(DOCX)

**S2 Table. Cox regression analyses assessing risk factors for OS by receptor status (TNBC vs. non-TNBC).** HR: Hazard Ratio assessing risk of death for 1-unit increase, or for specific category vs reference; CI: Confidence Interval; P: p-value testing HR = 1 from Wald Test. N/A: not applicable due to the small number of events in a group. Note: For a variable with missing value, we clarified the evaluable number of participants and events after excluding the missing values.
(DOCX)

## Acknowledgments

None.

## Author contributions

**Conceptualization:** Raksha M. Narasimhan, Ifeanyichukwu Ogobuiro.

**Data curation:** Raksha M. Narasimhan, Ifeanyichukwu Ogobuiro, Cristiane Takita.

**Formal analysis:** Xuesen Zhao, Sunwoo Han.

**Investigation:** Raksha M. Narasimhan, Xuesen Zhao.

**Methodology:** Raksha M. Narasimhan, Xuesen Zhao, Sunwoo Han.

**Software:** Xuesen Zhao, Sunwoo Han.

 

**Supervision:** Ifeanyichukwu Ogobuiro, Cristiane Takita, Crystal S. Taswell.

**Writing – original draft:** Raksha M. Narasimhan, Aren Singh Saini, Kayla Samimi.

**Writing – review & editing:** Raksha M. Narasimhan, Aren Singh Saini, Ifeanyichukwu Ogobuiro, Xuesen Zhao, Sunwoo Han, Crystal S. Taswell.

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
