## [Decision Letter · Decision Letter 0]

26 Mar 2026

PONE-D-26-01325Postmastectomy Radiotherapy in pN1 Breast Cancer: Survival Outcomes and Prognostic Factors From a Single-Institution CohortPLOS One

Dear Dr. Narasimhan,

Thank you for submitting your manuscript to PLOS ONE. After careful consideration, we feel that it has merit but does not fully meet PLOS ONE’s publication criteria as it currently stands. Therefore, we invite you to submit a revised version of the manuscript that addresses the points raised during the review process.

**ACADEMIC EDITOR: Major Revision**

We look forward to receiving your revised manuscript.

Kind regards,

Satyajeet Rath

Academic Editor

PLOS One

Journal Requirements:

“None”

3. In the online submission form you indicate that your data is not available for proprietary reasons and have provided a contact point for accessing this data. Please note that your current contact point is a co-author on this manuscript. According to our Data Policy, the contact point must not be an author on the manuscript and must be an institutional contact, ideally not an individual. Please revise your data statement to a non-author institutional point of contact, such as a data access or ethics committee, and send this to us via return email. Please also include contact information for the third party organization, and please include the full citation of where the data can be found.

4. We notice that your supplementary tables are included in the manuscript file. Please remove them and upload them with the file type 'Supporting Information'. Please ensure that each Supporting Information file has a legend listed in the manuscript after the references list.

Reviewers' comments:

Reviewer's Responses to Questions

**Comments to the Author**

1. Is the manuscript technically sound, and do the data support the conclusions?

Reviewer #1: Yes

Reviewer #2: Partly

Reviewer #3: No

2. Has the statistical analysis been performed appropriately and rigorously? 

Reviewer #1: Yes

Reviewer #2: No

Reviewer #3: N/A

3. Have the authors made all data underlying the findings in their manuscript fully available?

Reviewer #1: Yes

Reviewer #2: Yes

Reviewer #3: Yes

4. Is the manuscript presented in an intelligible fashion and written in standard English?

Reviewer #1: Yes

Reviewer #2: Yes

Reviewer #3: Yes

5. Review Comments to the Author

Reviewer #1: This is an automated report for PONE-D-26-01325. This report was solicited by the PLOS One editorial team and provided by ScreenIT.

ScreenIT is an independent group of scientists developing automated tools that analyze academic papers. A set of automated tools screened your submitted manuscript and provided the report below. Each tool was created by your academic colleagues with the goal of helping authors. The tools look for factors that are important for transparency, rigor and reproducibility, and we hope that the report might help you to improve reporting in your manuscript. Within the report you will find links to more information about the items that the tools check. These links include helpful papers, websites, or videos that explain why the item is important. While our screening tools aim to improve and maintain quality standards they may, on occasion, miss nuances specific to your study type or flag something incorrectly. Each tool has limitations that are described on the ScreenIT website. The tools screen the main file for the paper; they are not able to screen supplements stored in separate files. Please note that the Academic Editor had access to these comments while making a decision on your manuscript. The Academic Editor may ask that issues flagged in this report be addressed. If you would like to provide feedback on the ScreenIT tool, please email the team at ScreenIt@bih-charite.de. If you have questions or concerns about the review process, please contact the PLOS One office at plosone@plos.org.

Reviewer #2: Information in tables of results section should be explained in texts as well.

The statistical analysis should be revisited. if only patient with mastectomy between 2016 to 2022 are taken then how the IQR upper range of follow up be 156 months in march 2026? The no. of patients between different demographic parameters are different. This needs explanation. e.g. - cancer type N=49, No of positive nodes N=51.and similarly there are different event rates for different parameters as well. This needs explanation.

in line 190-191 it is inappropriate to say tending to wards significant with wide CI & P=0.263. Such loose inferences are there in other places in results section as well.

in a study where patients included are patients with mastectomy in 2016 to 2022 the graphs of Kaplan Meier are reaching upto 288 months of follow up. in 2026 maximum time can be around 120 months.

Overall the data needs to be reverified and statistical analysis needs to be revisited with complete overhauling of the results section of the manuscript.

Reviewer #3: I extend my sincere appreciation to the Editor for the opportunity to review this manuscript and to the authors for their thoughtful submission. The study presents a descriptive, retrospective evaluation of two cohorts of breast cancer patients treated with mastectomy and axillary surgery at a single center—one cohort receiving post-mastectomy radiotherapy (PMRT) and the other not. The primary objective was to compare survival outcomes between cohorts. The authors provide a comprehensive characterization of the study population, encompassing clinical and pathological features, and investigate associations between clinical/therapeutic variables and survival. Their detailed survival analyses for both cohorts highlight select clinical characteristics and reveal no significant differences. These findings may inform institutional understanding of the studied population and guide adaptations to local management protocols.

Nevertheless, the retrospective, descriptive methodology, coupled with modest sample sizes in each cohort, limits the generalizability of findings to diverse populations, thereby diminishing their wider applicability. Moreover, the manuscript recapitulates well-established prognostic factors without introducing “novel” insights or contributions specific to this population. In light of these major constraints, I cannot recommend acceptance for publication in this journal.

6. PLOS authors have the option to publish the peer review history of their article (what does this mean?). If published, this will include your full peer review and any attached files.

Reviewer #1: No

Reviewer #2: No

Reviewer #3: No

---

## [Author Response · Author response to Decision Letter 1]

14 Apr 2026

Response to Reviewers: Postmastectomy Radiotherapy in pN1 Breast Cancer: Survival Outcomes and Prognostic Factors From a Single-Institution Cohort

Journal Requirements:

Response: The manuscript has been revised to meet PLOS ONE’s style requirements.

2. Please remove any funding-related text from the manuscript and let us know how you would like to update your Funding Statement. Currently, your Funding Statement reads as follows: “The author(s) received no specific funding for this work.” Please include your amended statements within your cover letter; we will change the online submission form on your behalf.

Response: All funding-related text has been removed from the manuscript. The amended funding statement is included in our revised cover letter.

3. Please revise your data statement to a non-author institutional point of contact, such as a data access or ethics committee, and send this to us via return email. Please also include contact information for the third-party organization, and please include the full citation of where the data can be found.

Response: We changed the contact point in the submission form to the following:

“Data supporting the findings of this study are not publicly available due to institutional and ethical restrictions related to patient confidentiality. The data are available upon reasonable request from the University of Miami Human Subjects Research Office, which serves as the institutional oversight body for this study. Requests for data access may be sent to:

Human Subjects Research Office

University of Miami

Email: hsro@miami.edu

Phone: 305-243-3195

Website: www.hsro.miami.edu

Study reference:

HSRO Study Number: 20090856

Study Title: Collection of Characteristics and Outcome of Patients Treated for Breast Cancer by Physicians in the Department of Radiation Oncology

IRB Approval Date: December 1, 2009

Data access is subject to institutional review and approval and may require a data use agreement in accordance with University of Miami policies.”

4. We notice that your supplementary tables are included in the manuscript file. Please remove them and upload them with the file type 'Supporting Information'. Please ensure that each Supporting Information file has a legend listed in the manuscript after the references list.

Response: The requested changes have been made.

Response: Not applicable.

Comments to the Author:

We thank the reviewers and editorial team for their careful evaluation of our manuscript. We have addressed the concerns raised and outline our responses below.

1. Is the manuscript technically sound, and do the data support the conclusions?

Reviewer #1: Yes

Reviewer #2: Partly

Reviewer #3: No

Response: We appreciate the reviewers’ assessment. In response, we carefully revised the manuscript to ensure that our conclusions are appropriately supported by the data and not overstated. Specifically, we refined the language in the Results and Discussion sections to avoid overinterpretation of non-significant findings and to more clearly align our results with our conclusions. We also strengthened the Discussion to more explicitly acknowledge the limitations of our study, including retrospective single-institution design and modest sample size, which may affect generalizability. We also clarified aspects of the dataset that were previously vague, including handling of missing data and variable-specific sample sizes. We hope these revisions address the concerns raised and reinforce that the study is methodologically sound, with appropriately grounded conclusions.

2. Has the statistical analysis been performed appropriately and rigorously?

Reviewer #1: Yes

Reviewer #2: No

Reviewer #3: N/A

Response: We thank Reviewer #2 for highlighting their concerns about our statistical analysis. We carefully revisited all analyses to ensure accuracy and internal consistency. Of note, we corrected the study timeframe to distinguish between the years of mastectomy (2016-2022) and the full range of diagnosis and follow-up (1999-2023), which resolves the previously noted inconsistencies in follow-up duration and Kaplan-Meier estimates. We also revised the Results section to more clearly describe the statistical findings presented in the tables, ensuring that key results are interpretable within the text and that readers do not have to rely on tables alone to make sense of our findings. In addition, we clarified discrepancies in sample sizes across variables by explicitly noting missing data. We also reported evaluable denominators for different variables. We removed imprecise language such as “trending toward significance” and replaced these phrases with more accurate and specific interpretations of statistical results. We strongly believe these revisions substantially improve the rigor, clarity, and transparency of the statistical analysis.

3. Have the authors made all data underlying the findings in their manuscript fully available?

Reviewer #1: Yes

Reviewer #2: Yes

Reviewer #3: Yes

Response: We appreciate the reviewers’ confirmation of our data. As previously noted, we have further clarified our Data Availability Statement to include a non-author institutional point of contact (University of Miami Human Subjects Research Office), along with full contact information and study citation.

4. Is the manuscript presented in an intelligible fashion and written in standard English?

Reviewer #1: Yes

Reviewer #2: Yes

Reviewer #3: Yes

Response: We thank the reviewers for their positive assessment. While making revisions, we have further edited the manuscript for clarity and flow, particularly in the Results and Discussion sections.

Reviewer Comments to the Author

Reviewer #1: This is an automated report for PONE-D-26-01325. This report was solicited by the PLOS One editorial team and provided by ScreenIT.

ScreenIT is an independent group of scientists developing automated tools that analyze academic papers. A set of automated tools screened your submitted manuscript and provided the report below.

Responses:

1. Rigor

a. Randomization: Not applicable (retrospective cohort study)

b. Blinding: Not applicable (retrospective cohort study)

c. Power Analysis: A formal power/sample size calculation was not performed, as this study was retrospective in design. The sample size was determined by including all eligible patients who met inclusion criteria during the study period.

2. Transparency

a. Open Code: No custom code was used for the analysis. Statistical analyses were performed using standard software (SAS version 9.4).

b. Open Data: Due to IRB restrictions and patient privacy concerns, the data is not available to the public. Data is stored in a secure institutional repository and may be accessed upon reasonable request to the corresponding author in accordance with institutional policies.

Reviewer #2:

1. Information in tables of results section should be explained in texts as well.

Response: We have revised the Results section to more explicitly describe and interpret the key findings presented in Tables 1-3. We summarized descriptions of baseline cohort characteristics and highlighted the most clinically relevant associations from the univariable Cox regression analyses to improve clarity and ensure that the reader may understand the main findings from the manuscript itself rather than having to rely on the tables.

2. The statistical analysis should be revisited. If only patient with mastectomy between 2016 to 2022 are taken then how the IQR upper range of follow up be 156 months in March 2026?

Response: Thank you for pointing out this error. We fixed this mistake by correcting the date range to 1999-2023, which represents the earliest date of diagnosis to the latest date of follow-up. The 2016-2022 time frame corresponded the range of years during which participants underwent mastectomy.

3. The no. of patients between different demographic parameters are different. This needs explanation. e.g. - cancer type N=49, No of positive nodes N=51 and similarly there are different event rates for different parameters as well. This needs explanation.

Response: Thank you for the comment. This observed discrepancy is due to variable-specific missing values. To further clarify this within the manuscript, we added note of "For a variable with missing value, we clarified the evaluable number of participants and events after excluding the missing values." to Tables 2 and 3 as well as Supplemental Table 1 and 2.

4. In line 190-191 it is inappropriate to say tending towards significant with wide CI & P=0.263. Such loose inferences are there in other places in results section as well.

Response: We changed the wording to “not significant differences between the PMRT and non-PMRT groups” in line 190-191. This was the only place we encountered such an issue.

5. In a study where patients included are patients with mastectomy in 2016 to 2022 the graphs of Kaplan Meier are reaching up to 288 months of follow up. in 2026 maximum time can be around 120 months.

Response: Please see our response to comment #1. With the corrected timeframe of 1999-2023, the maximum follow-up time is consistent with 288 months.

Reviewer #3: I extend my sincere appreciation to the Editor for the opportunity to review this manuscript and to the authors for their thoughtful submission. The study presents a descriptive, retrospective evaluation of two cohorts of breast cancer patients treated with mastectomy and axillary surgery at a single center—one cohort receiving post-mastectomy radiotherapy (PMRT) and the other not. The primary objective was to compare survival outcomes between cohorts. The authors provide a comprehensive characterization of the study population, encompassing clinical and pathological features, and investigate associations between clinical/therapeutic variables and survival. Their detailed survival analyses for both cohorts highlight select clinical characteristics and reveal no significant differences. These findings may inform institutional understanding of the studied population and guide adaptations to local management protocols.

Nevertheless, the retrospective, descriptive methodology, coupled with modest sample sizes in each cohort, limits the generalizability of findings to diverse populations, thereby diminishing their wider applicability. Moreover, the manuscript recapitulates well-established prognostic factors without introducing “novel” insights or contributions specific to this population. In light of these major constraints, I cannot recommend acceptance for publication in this journal.

Response: We sincerely thank the reviewer for their thoughtful and detailed evaluation of our manuscript and for highlighting these important considerations. We agree that the retrospective single-institution design and modest sample size limit generalizability and statistical power. We have revised the Discussion to more explicitly acknowledge these limitations. We have also refined the framing of our study to better contextualize its contribution within the existing literature. While our findings are consistent with previously established prognostic factors and do not introduce novel predictors, we emphasize that they provide contemporary validation in the real-world modern era of systemic therapy, where the absolute benefit of PMRT in pN1 disease remains under ongoing debate. We have also adjusted our limitations and conclusions sections to avoid overinterpretation and jumping to false conclusions. We hope these revisions clarify the value of our study in its contributions to contemporary, individualized decision-making in this clinically heterogeneous population.

---

## [Decision Letter · Decision Letter 1]

12 May 2026

Postmastectomy Radiotherapy in pN1 Breast Cancer: Survival Outcomes and Prognostic Factors From a Single-Institution Cohort

PONE-D-26-01325R1

Dear Dr. Narasimhan,

We’re pleased to inform you that your manuscript has been judged scientifically suitable for publication and will be formally accepted for publication once it meets all outstanding technical requirements.

Kind regards,

Satyajeet Rath

Academic Editor

PLOS One

Additional Editor Comments (optional):

Reviewers' comments:

Reviewer's Responses to Questions

**Comments to the Author**

1. If the authors have adequately addressed your comments raised in a previous round of review and you feel that this manuscript is now acceptable for publication, you may indicate that here to bypass the “Comments to the Author” section, enter your conflict of interest statement in the “Confidential to Editor” section, and submit your "Accept" recommendation.

Reviewer #2: All comments have been addressed

Reviewer #3: (No Response)

2. Is the manuscript technically sound, and do the data support the conclusions?

Reviewer #2: Yes

Reviewer #3: No

3. Has the statistical analysis been performed appropriately and rigorously? 

Reviewer #2: Yes

Reviewer #3: N/A

4. Have the authors made all data underlying the findings in their manuscript fully available?

Reviewer #2: Yes

Reviewer #3: Yes

5. Is the manuscript presented in an intelligible fashion and written in standard English?

Reviewer #2: Yes

Reviewer #3: Yes

6. Review Comments to the Author

Reviewer #2: Authors have incorporated the changes as suggested and have made the required changes in the manuscript.

Reviewer #3: I thank the authors for the review and the revisions made to the manuscript; however, I believe the primary limitation of the manuscript is the sample size of the study, which is insufficiently representative to permit generalization of the conclusions and renders the study's relevance somewhat difficult to accept for publication in this journal

7. PLOS authors have the option to publish the peer review history of their article (what does this mean?). If published, this will include your full peer review and any attached files.

Reviewer #2: No

Reviewer #3: No

---

## [Editor Report · Acceptance letter]

PONE-D-26-01325R1

PLOS One

Dear Dr. Narasimhan,

I'm pleased to inform you that your manuscript has been deemed suitable for publication in PLOS One. Congratulations! Your manuscript is now being handed over to our production team.

Kind regards,

on behalf of

Dr. Satyajeet Rath

Academic Editor

PLOS One